# Understanding and Improving Robustness of Vision Transformers through Patch-based Negative Augmentation

**Yao Qin    Chiyuan Zhang    Ting Chen**

**Balaji Lakshminarayanan    Alex Beutel    Xuezhi Wang**

Google Research

## Abstract

We investigate the robustness of vision transformers (ViTs) through the lens of their special patch-based architectural structure, i.e., they process an image as a sequence of image patches. We find that ViTs are surprisingly insensitive to patch-based transformations, even when the transformation largely destroys the original semantics and makes the image unrecognizable by humans. This indicates that ViTs heavily use features that survived such transformations but are generally not indicative of the semantic class to humans. Further investigations show that these features are useful but *non-robust*, as ViTs trained on them can achieve high in-distribution accuracy, but break down under distribution shifts. From this understanding, we ask: can training the model to rely less on these features improve ViT robustness and out-of-distribution performance? We use the images transformed with our patch-based operations as negatively augmented views and offer losses to regularize the training away from using non-robust features. This is a *complementary* view to existing research that mostly focuses on augmenting inputs with semantic-preserving transformations to enforce models' invariance. We show that patch-based negative augmentation consistently improves robustness of ViTs on ImageNet based robustness benchmarks across 20+ different experimental settings. Furthermore, we find our patch-based negative augmentation are complementary to traditional (positive) data augmentation techniques and batch-based negative examples in contrastive learning.

## 1 Introduction

Building vision models that are robust, i.e., that maintain accuracy even on unexpected and out-of-distribution images, is increasingly a requirement to trusting vision models and a strong benchmark for progress in the field. Recently, Vision Transformers (ViTs, Dosovitskiy et al. (2021)) sparked great interest in the literature, as a radically new model architecture offering significant accuracy improvements and with hope of new robustness benefits. Over the past decade, there has been extensive work on understanding the robustness of convolution-based neural architectures, as the dominant design for visual tasks; researchers have explored adversarial robustness (Szegedy et al., 2013), domain generalization (Xiao et al., 2021; Khani & Liang, 2021), feature biases (Brendel & Bethge, 2019; Geirhos et al., 2018; Hermann et al., 2020). As a result, with the new promise of vision transformers, it is critical to understand *their* properties and in particular their robustness. Recent early studies (Naseer et al., 2021; Paul & Chen, 2021; Bhojanapalli et al., 2021) have found ViTs be more robust than ConvNets in some scenarios, with the hypothesis that the non-local attention based interactions enabled ViTs to capture more global and semantic features. In contrast, we add to this

36th Conference on Neural Information Processing Systems (NeurIPS 2022).

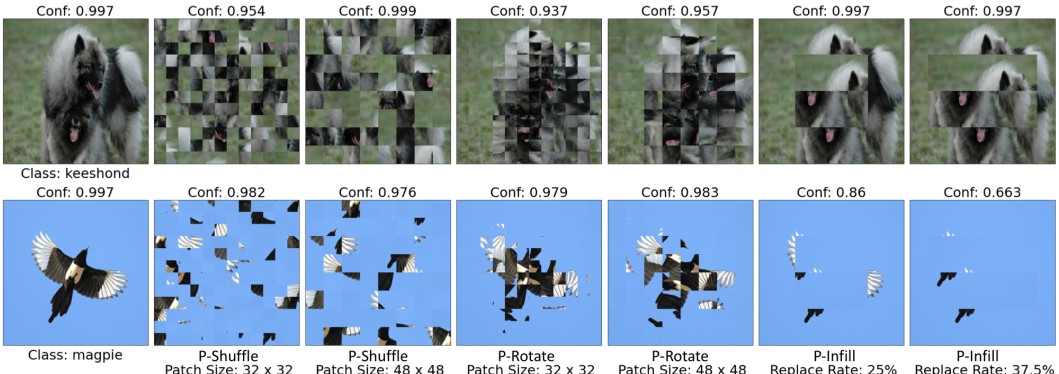

Figure 1: Patch-based transformations largely destroy images to be unrecognizable to humans whereas ViT recognizes them as the original class (e.g., keeshond or magpie) with high confidence. Visualization of patch-based transformations. On the top of each image, we display the predicted confidence score of ViT-B/16 pretrained on ImageNet-21k and finetuned on ImageNet-1k.

line of research showing a different side of the challenge: we find that ViTs are still vulnerable to relying on non-robust features, and propose an algorithm to mitigate it for better out-of-distribution performance.

Specifically, we first conduct a study on the robustness properties of ViTs and show that ViTs still rely on specific non-robust features. We start with the architectural traits of ViTs – ViTs operate on non-overlapping image patches and allow long range interaction between patches even in lower layers. It is hypothesized in recent studies (Naseer et al., 2021; Paul & Chen, 2021; Bhojanapalli et al., 2021) that the non-local attention based interactions contribute to better robustness of ViTs than ConvNets. To study the ability of ViTs to integrate global semantics structures across patches, we design and apply patch-based image transformations, such as random patch rotation, shuffling, and background-infilling (Figure 1). Those transformations destroy the spatial relationship between patches and corrupted the global semantics, and the resultant images are often visually unrecognizable. However, we find that ViTs are surprisingly insensitive to these transformations and can make highly accurate predictions on these transformed images. This suggests that ViTs use features that survive such transformations but are generally not indicative of the semantic class to humans. Going one step further, we find that those features are useful but not robust, as ViTs trained on them achieved high in-distribution accuracy, but suffered significantly on robustness benchmarks.

With this understanding of ViTs' reliance on non-robust features captured by patch-based transformations, we aim to mitigate this shortcoming by answering the following questions: (a) how can we train ViTs to not rely on such features? and (b) will reducing reliance on such features meaningfully improve out-of-distribution performance and not sacrifice in-distribution accuracy? A majority of past robust training algorithms encourage the smoothness of model predictions on augmented images with semantic *preserving* transformations (Hendrycks et al., 2020; Cubuk et al., 2019). However, the patch-based transformations deliberately destroy the semantic meaning and only leave non-robust features. Taking inspiration from recent research on generative modeling (Sinha et al., 2020), we propose a family of robust training algorithms based on *patch-based negative augmentations* that regularize the training from relying on non-robust features surviving patch-based transformations. Through extensive evaluation on a wide set of ImageNet-based benchmarks, we find that our methods consistently improve the robustness of the trained ViTs. Furthermore, our patch-based negative augmentation can be combined with the traditional (positive) data augmentation to boost the performance further. With this we get a more complete picture: training models both to be insensitive to spurious changes (as in positive augmentation) but also to not rely on non-robust features (as in negative augmentation) together can meaningfully improve robustness of ViTs.

Our key contributions are as follows:

- **Understanding Non-Robust Features in ViT:** We show that ViTs heavily rely on non-robust features surviving patch-based transformations but are not indicative of the semantic classes to humans.

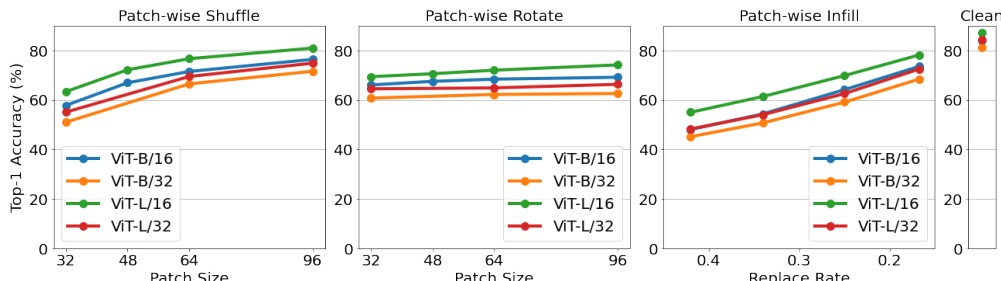

Figure 2: ViTs can rely on features surviving patch-based transformations to maintain a high accuracy, even after images have been heavily transformed to be largely unrecognizable. Top-1 accuracy of ViT models when tested on patch-based transformed images using the semantic class of the corresponding clean image as ground-truth. The test accuracy on ImageNet-1k validation set is shown on the right. All ViT models are pre-trained on ImageNet-21k and fine-tuned on ImageNet-1k.

- **Modeling:** We propose a set of patch-based operations as negatively augmented views, complementary to existing works that focus on semantic-preserving ("positive") augmentations, to regularize the training away from using these specific non-robust features;

- **Improved Robustness of ViT:** We show across 20+ experimental settings that our proposed patch-based negative augmented views can consistently improve ViT's robustness and complementary to "positive" augmentation techniques as well as batch-based negative examples in contrastive learning.

## 2 Preliminaries

**Vision Transformers**  Vision transformers (Dosovitskiy et al., 2021) are a family of architectures adapted from Transformers in natural language processing (Vaswani et al., 2017), which directly process visual tokens constructed from image patch embeddings. To construct visual tokens, ViT (Dosovitskiy et al., 2021) first splits an image into a grid of patches. Each patch is linearly projected into a hidden representation vector, and combined with a positional embedding. A learnable class token is also added. Transformers (Vaswani et al., 2017) are then directly applied on this set of visual and class token embeddings as if they are word embeddings in the original Transformer formulation. Finally, a linear projection of class token is used to calculate the class probability.

**Model Variants**  We consider ViT models pretrained on either ILSVRC-2012 ImageNet-1k, with $\sim$1.3 million images or ImageNet-21k, with $\sim$14 million images (Russakovsky et al., 2015). All models are fine-tuned on ImageNet-1k dataset. We adopt the notations used in (Dosovitskiy et al., 2021) to denote model size and input patch size. For example, ViT-B/16 denotes the "Base" model variant with input patch size $16 \times 16$.

**Robustness Benchmarks**  To evaluate models' robustness, we mainly focus on three ImageNet-based robustness benchmarks, ImageNet-A (Hendrycks et al., 2019), ImageNet-C (Hendrycks & Dietterich, 2019a), ImageNet-R (Hendrycks et al., 2021). Specifically, ImageNet-A contains challenging natural images from a distribution unlike ImageNet training distribution, which can easily fool models to make a misclassification. ImageNet-C consists of 19 types of corruptions that are frequently encountered in natural images and each corruption has 5 levels of severity. It is widely used to measure models' robustness under distributional shift. ImageNet-R is composed of images obtained by artistic rendition of ImageNet classes, e.g., cartoons, and is widely used to evaluate model's robustness on out-of-distribution data.

## 3 Understanding the Robustness of Vision Transformers

Recent works (Naseer et al., 2021; Paul & Chen, 2021; Bhojanapalli et al., 2021) have shown that vision transformers achieve better robustness compared to standard convolutional networks. One explanation is that the attention mechanism can capture better global structures. To investigate if ViT has successfully taken advantage of the long range interactions between patches, we design a series of patch-based transformations which significantly destroys the global structure of images. The patch-based transformations (see Fig. 1) are:

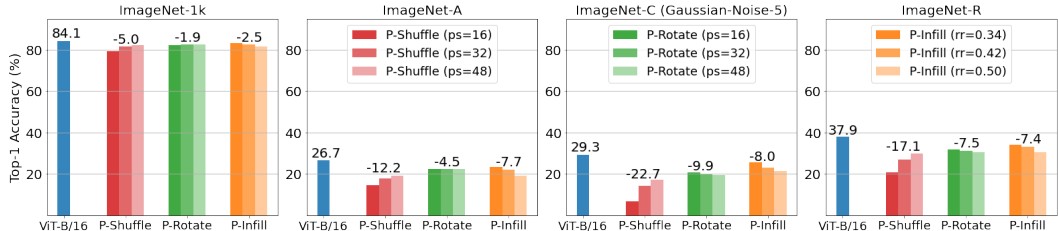

Figure 3: Features preserved in patch-based transformations are useful but non-robust as training ViT on them impedes robustness. Top-1 Accuracy (%) on ImageNet-1k validation set and ImageNet robustness datasets: ImageNet-A, ImageNet-C, ImageNet-R. The baseline model is ViT-B/16 in (Dosovitskiy et al., 2021) trained on original images. Other models are trained on patch-based transformed images, e.g., "P-Shuffle" stands for a ViT-B/16 model trained on patch-based shuffled images. Numbers above the bars are either accuracy (e.g., ViT-B/16) or the *max* accuracy difference between each model family and the baseline ViT-B/16. The patch size in P-Shuffle and P-Rotate and replacement ratio in P-Infill is denoted by "ps" and "rr" respectively.

- **Patch-based Shuffle (P-Shuffle)**: we randomly shuffle the input image patches to change their positions[1].
- **Patch-based Rotate (P-Rotate)**: we randomly select a rotation degree from the set $\Omega = \{0°, 90°, 180°, 270°\}$ and rotate each image patch independently.
- **Patch-based Infill (P-Infill)**: we replace the image patches in the center region of an image with the patches on the image boundary[2].

Each patch-based transformation is performed to a single image. We make sure the patch size of our patch-based transformation is a multiple of the input image patch of ViT so that the content within each patch is well-maintained. For P-infill, we use "replace rate" to denote the ratio of replaced patches in the center over the total number of patches in an image. Examples of transformed images are shown in Fig. 1 (see Appendix E for more examples). In most cases, it is challenging to recognize the semantic classes after those transformations.

***Do ViTs rely on features not indicative of the semantic classes to humans?*** To validate if ViTs behave similarly as humans on these patch-based transformed images, we *evaluate* ViT models (Dosovitskiy et al., 2021) on these patch-based transformed image. Specifically, we apply each patch-based transformation to ImageNet-1k validation set and report the test accuracy of each ViT on the transformed images. The test accuracy is computed by using the semantic class of the corresponding original image as the ground-truth. As shown in Figure 2, the accuracies achieved by ViTs are significantly higher than random guessing (0.1%). In addition, as shown in Figure 1, ViT gives these patch-based transformed images a highly confident prediction even when the transformation largely destroys the semantics and make the image unrecognizable by humans.[3] This strongly indicates that ViT models heavily rely on the features that survive these transformations to make a prediction. However, these features are not indicative of the semantic class to humans.

***Do features preserved in patch-based transformations impede robustness?*** Taking one step further, we want to know if the features preserved by simple patch-based transformations, which are not indicative of the semantic class to humans, result in robustness issues. To this end, we train a vision transformer, e.g., ViT-B/16, on patch-based transformed images with original semantic class assigned as their ground-truth. Note that all the training images are patch-based transformed images. In this way, we force the model to fully exploit the features preserved in patch-based transformations. In addition, we reuse all the training details and hyperparameters in (Dosovitskiy et al., 2021) to make sure the "only" difference between our models and the baseline ViT-B/16 (Dosovitskiy et al., 2021) is the training images. Then, we test the model on ImageNet-1k validation set and three robustness benchmarks, ImageNet-A, ImageNet-C and ImageNet-R *without any transformation*.

---

[1]P-Shuffle is equivalent to shuffling the position embeddings.

[2]For example, given an image with size $384 \times 384$, input patch size is $16 \times 16$ and replace rate 0.25, we in total have 576 patches $\boldsymbol{x}_{i,j}$, where $i$ and $j$ denotes the row and column index and $1 \leq i, j \leq 24$. The patches in the center $\boldsymbol{x}_{m,n}, 7 \leq m, n \leq 18$ are replaced by the remaining patches.

[3]Similar patterns are observed when ViT models are pretrained on ImageNet-1k.

First, we can observe that for the baseline ViT-B/16, compared to in-distribution accuracy, all the out-of-distribution accuracies have suffered from a significant drop (the 4 blue bars in Figure 3). This trend has been observed both for ViTs and convolution-based networks (Paul & Chen, 2021; Zhai et al., 2021). Second, if we compare the accuracy between the baseline model and models trained on patch-based transformations (i.e., the difference between the blue bar and one of the red/green/orange bars in Figure 3), we find that ViTs' in-distribution accuracy drops only slightly, but the robustness drop is significant when models are trained on these patch-based transformations. Take P-Shuffle as an example, the model trained on patch-based shuffled images can still achieve 79.1% accuracy on ImageNet-1k, only 5 percentage point (pp) drop in in-distribution accuracy. In contrast, the accuracy drop on robustness datasets is much more significant, e.g., 17pp on ImageNet-R. The deterioration rate in robustness is close to 50% of the baseline ViT-B/16. This strongly suggests that the features preserved in patch-based transformations are sufficient for high-accurate in-distribution prediction but are not robust under distributional shifts.

Taking the above results together, we conclude that even though ViTs are shown to be more robust than ConvNets in previous studies (Naseer et al., 2021; Paul & Chen, 2021), they still heavily rely on features that are not indicative of the semantic classes to humans. These features, captured by patch-based transformations, are **useful** but **non-robust**, as ViTs trained on them achieve high in-distribution accuracy but suffer significantly on robustness benchmarks.

With this knowledge we now ask: *based on these understandings, can we train ViTs to not rely on such non-robust features? And if we do, will it improve their robustness?*

## 4 Improving the Robustness of Vision Transformers

Based on the key observations that the patch-based transformations encode features that contribute to the non-robustness of ViTs, we propose a *negative augmentation* procedure to regularize ViTs from relying on such features. To this end, we use the images transformed with our patch-based operations as negatively augmented views. Then we design negative loss regularizations to prevent the model from using those non-robust features preserved in patch-based transformations.

Specifically, given a clean image $x$, we generate its negative view, denoted as $\tilde{x}$, by applying a patch-based transformation to $x$. We call it *negative* augmentation, in contrast with the standard (positive) augmentation that are semantic preserving. Let $\mathcal{L}_{ce}(\mathbb{B}; \boldsymbol{\theta}) = -\frac{1}{|\mathbb{B}|} \sum_{(x,y) \in \mathbb{B}} \boldsymbol{y} \log \mathrm{softmax}(f(x; \boldsymbol{\theta}))$ represent the cross-entropy loss function used to train a vision transformer with parameters $\boldsymbol{\theta}$, where $\mathbb{B}$ is a minibatch of clean examples, and $\boldsymbol{y}$ denotes the ground-truth label. The loss on negative views $\mathcal{L}_{neg}(\mathbb{B}, \tilde{\mathbb{B}}; \boldsymbol{\theta})$ can be easily added to the cross-entropy loss $\mathcal{L}_{ce}(\mathbb{B}; \boldsymbol{\theta})$ via

$$\mathcal{L}_{ce}(\mathbb{B}; \boldsymbol{\theta}) + \lambda \cdot \mathcal{L}_{neg}(\mathbb{B}, \tilde{\mathbb{B}}; \boldsymbol{\theta}), \tag{1}$$

where $\lambda$ is a coefficient balancing the importance between clean training data as well as patch-based negative augmentation. Below, we introduce uniform loss and $\ell_2$ loss to leverage negative views.

**Uniform loss** Many existing data augmentation techniques (Cubuk et al., 2019, 2020; Hendrycks et al., 2020) use one-hot labels for *semantic-preserving* augmented data to enforce the invariance of the model prediction. In contrast, the semantic classes of our generated patch-based negative augmented data are visually unrecognizable, as shown in Figure 1. Therefore, we propose to use uniform labels instead for those negative augmentations. Specifically, the loss function on negative views that we optimize at each training step can be formulated as:

$$\mathcal{L}_{neg}(\mathbb{B}, \tilde{\mathbb{B}}; \boldsymbol{\theta}) = -\frac{1}{|\tilde{\mathbb{B}}|} \sum_{(\tilde{x}, \tilde{y}) \in \tilde{\mathbb{B}}} \tilde{\boldsymbol{y}} \log \mathrm{softmax}(f(\tilde{x}; \boldsymbol{\theta})), \tag{2}$$

where $\tilde{\boldsymbol{y}}$ denotes the uniform distribution: $\tilde{y}_k = \frac{1}{K}$ where $K$ is the total number of classes. $f(x; \boldsymbol{\theta})$ denotes the function mapping the input image into the logit space.

$\ell_2$ **Loss** An alternative to pre-assuming labels for negative augmentation is to add the constraints on the logit space (or the space of predicted probability). Inspired by existing work (Kannan et al., 2018; Zhang et al., 2019; Hendrycks et al., 2020) which provides an extra regularization term encouraging similar logits between clean and "positive" augmented counterparts, we instead encourage the logits of clean examples and their corresponding negative augmentations to be far away. In this way, we prevent the model from relying on the non-robust features preserved in negative views.

Table 1: Top-1 accuracies for ViT models pre-trained and fine-tuned on ImageNet-1k with or without the proposed negative augmentation. Best results highlighted in **bold**.

| Model | ImageNet-1k | ImageNet-A | ImageNet-C | ImageNet-R |
|---|---|---|---|---|
| ViT-B/32 (Dosovitskiy et al., 2021) | 72.5 | 3.7 | 46.7 | 19.4 |
| + P-Rotate / Uniform | 72.9 (+0.4) | 3.7 (+0.0) | 47.9 (+1.2) | 19.9 (+0.5) |
| + P-Rotate / L2 | **73.2 (+0.7)** | **3.8 (+0.1)** | **48.4 (+1.7)** | **20.4 (+1.0)** |
| ViT-B/16 (Dosovitskiy et al., 2021) | 77.6 | 6.7 | 50.8 | 20.3 |
| + P-Rotate / Uniform | **78.2 (+0.6)** | **7.0 (+0.3)** | **52.4 (+1.6)** | **21.4 (+1.1)** |
| + P-Rotate / L2 | 77.8 (+0.2) | 6.7 (+0.0) | 51.6 (+0.8) | 21.0 (+0.7) |

Specifically, we maximize the $\ell_2$ distance between the predicted probability of clean examples and their corresponding negative views. The loss on negative views, therefore, can be formulated as:

$$\mathcal{L}_{neg}(\mathbb{B}, \tilde{\mathbb{B}}; \boldsymbol{\theta}) = -\frac{1}{|\tilde{\mathbb{B}}|} \sum_{\boldsymbol{x} \in \mathbb{B}, \tilde{\boldsymbol{x}} \in \tilde{\mathbb{B}}} \|\mathrm{softmax}(f(\boldsymbol{x}; \boldsymbol{\theta})) - \mathrm{softmax}(f(\tilde{\boldsymbol{x}}; \boldsymbol{\theta}))\|_2. \tag{3}$$

Here the $\ell_2$ distance is computed over the predicted probability rather than the logits $f(\boldsymbol{x}; \boldsymbol{\theta})$ because empirically we observe that maximizing the difference of logits can cause numerical instability.

# 5 Experiments

**Experimental setup** We follow Dosovitskiy et al. (2021) to first pre-train all the models with image size $224 \times 224$ and then fine-tune the models with a higher resolution $384 \times 384$. We reuse all their training hyper-parameters, including batch size, weight decay, and training epochs (see Appendix B for details). For the extra loss coefficient $\lambda$ in Eqn. 1 we sweep it from the set $\{0.5, 1, 1.5\}$ and choose the model with the best hold-out validation performance. Please refer to Appendix C for the chosen hyperparameters for each model. We make sure our proposed models and our implemented baselines are trained with exactly the same settings for fair comparison.

## 5.1 Effective in improving robustness of vanilla ViTs

First, we apply our proposed patch-based transformations to a ViT model pre-trained and fine-tuned on ImageNet-1k. The extra loss regularization on negative views is used in both pre-training and fine-tuning stages to prevent the model from learning non-robust features preserved in patch-based transformations. We use "Transformation / Regularization" to denote a pair of patch-based negative augmentation and loss regularization. For examples, "P-Rotate / Uniform" means that we use P-Rotate to generate the negative views and use uniform loss to regularize the training. We display the results in Table 1, where we can clearly see that our proposed patch-based negative augmentation effectively improves the out-of-distribution robustness on ImageNet-C and ImageNet-R. Note the improvement is small over ImageNet-A when the baseline almost does not work, but it becomes more siginificant when the baseline improves (e.g., Table 5).

**Larger improvement on small-scale datasets** We also apply our patch-based negative augmentation to ViT-B/16 on CIFAR-100 (Krizhevsky, 2009) in the fine-tuning stage and find that it can also significantly improve the robustness on the corrupted dataset CIFAR-100-C (Hendrycks & Dietterich, 2019b), which includes 19 different corruption types with 5 different corruption severities. For example, P-Shuffle / Uniform can boost the in-distribution accuracy of ViT-B/16 on CIFAR-100 from 91.8% to 92.6% (+0.8) and the robustness accuracy on CIFAR-100-C can also be improved from 74.6% to 77.0% (+2.4), as shown in Table 10 in Appendix.

## 5.2 Complementary to traditional ("positive") data augmentation

To investigate if our proposed patch-based negative augmentation is complementary to traditional ("positive") data augmentation, we apply our patch-based negative transformation on top of traditional data augmentation: Rand-Augment (Cubuk et al., 2020), which is widely used in vision transformers (Touvron et al., 2021; Mao et al., 2021), and AugMix (Hendrycks et al., 2020), which is specifically proposed to improve models' robustness under distributional shift. Crucially, we follow (Hendrycks et al., 2020) to exclude transformations used in "positive" data augmentation which overlap with corruption types in ImageNet-C (Hendrycks & Dietterich, 2019a). Therefore, the set of transformations used in Rand-Augment and AugMix is disjoint with the corruptions in ImageNet-C.

When we combine "negative" and "positive" augmentation, the cross-entropy loss $\mathcal{L}_{ce}(\mathbb{B}^+; \boldsymbol{\theta})$ in Eqn. 1 is computed over "positive" examples $\mathbb{B}^+ = \{\boldsymbol{x}_1^+, \cdots, \boldsymbol{x}_N^+\}$ using either Rand-Augment or

Table 2: Patch-based negative augmentation is complementary to "positive" data augmentation. Top-1 accuracies for ViT-B/16 pre-trained and fine-tuned on ImageNet-1k using Rand-Augment (Cubuk et al., 2020) or AugMix (Hendrycks et al., 2020). The proposed negative augmentation is added on top of either positive augmentation. Best results are highlighted in **bold**.

| Model | ImageNet-1k | ImageNet-A | ImageNet-C | ImageNet-R |
|---|---|---|---|---|
| Rand-Augment (Cubuk et al., 2020) | 79.1 | 7.2 | 55.2 | 23.0 |
| + P-Shuffle / Uniform | 79.3 | 7.7 (+0.5) | 56.2 (+1.0) | 23.4 (+0.4) |
| + P-Rotate / Uniform | 79.3 | **8.1 (+0.9)** | 56.4 (+0.8) | 23.8 (+0.8) |
| + P-Infill / Uniform | 79.2 | 7.8 (+0.6) | 56.4 (+1.2) | **24.0 (+1.0)** |
| + P-Shuffle / L2 | 78.9 | 7.5 (+0.3) | 55.6 (+0.4) | 22.6 (-0.4) |
| + P-Rotate / L2 | 79.1 | 7.9 (+0.7) | **56.7 (+1.5)** | 23.8 (+0.8) |
| + P-Infill / L2 | 78.8 | 7.4 (+0.2) | 55.4 (+0.2) | 23.2 (+0.2) |
| AugMix (Hendrycks et al., 2020) | 78.8 | 7.7 | 57.8 | 24.9 |
| + P-Shuffle / Uniform | 79.2 | 8.0 (+0.3) | 58.6 (+0.8) | 25.7 (+0.8) |
| + P-Rotate / Uniform | 79.1 | 8.2 (+0.5) | 58.5 (+0.7) | 25.7 (+0.8) |
| + P-Infill / Uniform | 79.3 | **8.3 (+0.6)** | 58.4 (+0.6) | 25.7 (+0.8) |
| + P-Shuffle / L2 | 78.8 | 7.9 (+0.2) | 58.3 (+0.5) | 25.7 (+0.8) |
| + P-Rotate / L2 | 79.0 | **8.3 (+0.6)** | **58.8 (+1.0)** | **26.0 (+1.1)** |
| + P-Infill / L2 | 79.0 | 7.9 (+0.2) | 58.5 (+0.7) | 25.6 (+0.7) |

Table 3: Effects of patch-based negative augmentation on five different levels of corruption severities on ImageNet-C. Best results are highlighted in **bold**. See Table 12 in Appendix for full results.

| Model | ImageNet-C Corruption Severity Level | | | | |
|---|---|---|---|---|---|
| | 1 | 2 | 3 | 4 | 5 |
| Rand-Augment (Cubuk et al., 2020) | 70.4 | 63.7 | 57.9 | 48.2 | 36.1 |
| + P-Rotate / Uniform | **71.1 (+0.7)** | 64.6 (+0.9) | 59.0 (+1.1) | 50.0 (+1.8) | 37.6 (+1.5) |
| + P-Rotate / L2 | **71.1 (+0.7)** | **64.8 (+1.1)** | **59.5 (+1.6)** | **50.1 (+1.9)** | **37.8 (+1.7)** |
| AugMix (Hendrycks et al., 2020) | 71.4 | 65.2 | 60.5 | 51.9 | 40.2 |
| + P-Rotate / Uniform | 71.7 (+0.3) | 65.7 (+0.5) | 61.1 (+0.6) | 52.7 (+0.8) | 41.4 (+1.2) |
| + P-Rotate / L2 | **71.9 (+0.5)** | **66.0 (+0.8)** | **61.5 (+1.0)** | **52.9 (+1.0)** | **41.6 (+1.4)** |

AugMix. Meanwhile, the loss regularization on negative views in Eqn. 1 is computed over negatively transformed version of $x^+$. That is: for $\forall x^+ \in \mathbb{B}^+$, we apply our patch-based negative transformation to obtain its negative version and then use the negative example to compute the loss regularization $\mathcal{L}_{neg}$. The positive data augmentation is only used in pre-training stage as we observe it is slightly better than using them for both stages (Please see more detailed discussion in Appendix D.4) and Steiner et al. (2021) have the similar observation. Instead, we apply our negative augmentation in both stages, as it is the best design choice as discussed in Appendix D.3.

As shown in Table 2, we see that when our patch-based negative augmentations are applied to either Rand-Augment or AugMix, we can consistently improve the robustness of vision transformers across all three robustness benchmarks. This is particularly noteworthy as both Rand-Agument and AugMix are already designed to significantly improve the robustness of vision models. Yet, we see that patch-based negative augmentation provides *further* robustness benefits. This suggests that robustness of vision models was not adequately addressed by "positive" data augmentation and that patch-based negative augmentation is complementary to these traditional approaches.

To have a clear picture how negative augmentation improves robustness on data with different degrees of corruptions, we also display the top-1 accuracy on ImageNet-C with five different levels of corruption severity in Table 3. The general trend emerges: as the corruption level goes higher, the benefit of negative examples becomes larger. This suggests our proposed negative examples are especially useful when data is highly corrupted.

### 5.3 Complementary to batch-based negative examples in contrastive loss

In addition, we also investigate if our proposed patch-based negative augmentation can be incorporated into the traditional contrastive loss formulation and if they are complementary and provide additional value on top of the batch-based negative examples traditionally used (Oord et al., 2018; Chen et al., 2020a). We will first describe how to use contrastive loss as a regularization term and then introduce how to integrate patch-based negative augmentation into contrastive loss.

Table 4: Complementary to batch-based negative examples in contrastive loss. Top-1 accuracies of ViT-B/16 pretrained and fine-tuned on ImageNet-1k with and without P-Rotate.

| Model | ImageNet-1k | ImageNet-A | ImageNet-C | ImageNet-R |
|---|---|---|---|---|
| ViT-B/16 + Contrastive* | 78.7 | 8.1 | 53.5 | 22.8 |
| ViT-B/16 + P-Rotate / Contrastive | 78.9 | 8.6 (+0.5) | 54.1 (+0.6) | 23.6 (+0.8) |
| Rand-Augment + Contrastive* | 79.7 | 8.9 | 57.6 | 24.7 |
| Rand-Augment + P-Rotate / Contrastive | 79.9 | 9.4 (+0.5) | 58.4 (+0.8) | 25.4 (+0.7) |

Table 5: Patch-based negative augmentation is helpful even with large-scale pretraining. Top-1 accuracies of ViT-B/16 pretrained on ImageNet-21k and finetuned on ImageNet-1k. Mean±standard deviation are reported over 4 independent runs.

| Model | ImageNet-1k | ImageNet-A | ImageNet-C | ImageNet-R |
|---|---|---|---|---|
| Rand-Augment (Cubuk et al., 2020) | 84.4±0.0 | 28.7±0.2 | 67.2±0.0 | 38.7±0.1 |
| + P-Shuffle / Uniform | 84.5 | **29.9** | 67.7 | 38.9 |
| + P-Shuffle / L2 | 84.5±0.0 | 29.7±0.3 | **68.0±0.0** | **39.6±0.2** |

Following supervised contrastive loss proposed in (Khosla et al., 2020), for an example $x_i \in \mathbb{B}$, we create a positive set $\mathbb{P}_i \equiv \{x_j \in \mathbb{B}\setminus\{x_i\}|y_j = y_i\}$ with all the examples in the minibatch $\mathbb{B}$ sharing the same class as $x_i$. The anchor $x_i$ is excluded from its positive set $\mathbb{P}_i$. Next, all the examples in the minibatch $\mathbb{B}$ with a different class as $x_i$ are used as negative examples. Let the candidate set $\mathbb{Q}_i \equiv \mathbb{B}\setminus\{x_i\}$, the contrastive loss can be expressed as

$$\mathcal{L}_{cont}(\mathbb{B};\boldsymbol{\theta}) = -\frac{1}{|\mathbb{B}|}\sum_{x_i \in \mathbb{B}}\frac{1}{|\mathbb{P}_i|}\sum_{x_j \in \mathbb{P}_i}\log\frac{\exp(\text{sim}(x_i,x_j)/\tau)}{\sum_{x_k \in \mathbb{Q}_i}\exp(\text{sim}(x_i,x_k)/\tau)}, \quad (4)$$

where $\tau$ is the temperature and $\text{sim}(x_i,x_j) = \frac{g(x_i;\boldsymbol{\theta})^\top \cdot g(x_j;\boldsymbol{\theta})}{\|g(x_i;\boldsymbol{\theta})\|\|g(x_j;\boldsymbol{\theta})\|}$ computes the cosine similarity between $g(x_i;\boldsymbol{\theta})$ and $g(x_j;\boldsymbol{\theta})$, and $g(x;\boldsymbol{\theta})$ denotes the representation learned by the penultimate layer of the classifier. We do not use a learnable projection head as in contrastive representation learning (Chen et al., 2020a,b) to avoid extra network parameters. The contrastive loss $\mathcal{L}_{cont}$ is used as the regularization term $\mathcal{L}_{neg}$ in Eqn 1. We denote this stronger baseline as "Contrastive*".

To integrate our proposed patch-based negative examples into contrastive loss, we expand the negative set in contrastive loss, which now composed of two types of negative examples: 1) all the examples in the minibatch $\mathbb{B}$ with a different class as $x_i$, 2) the patch-based negatively transformed images $\tilde{x} \in \tilde{\mathbb{B}}$. Therefore, for each anchor $x_i$, we can in total have $2|\mathbb{B}| - |\mathbb{P}_i| - 1$ negative pairs, where $|\mathbb{B}|$ is the batch size and $|\mathbb{P}_i|$ is the cardinality of the positive set $\mathbb{P}_i$. Accordingly, the candidate set in Eqn 4 becomes $\mathbb{Q}_i \equiv \tilde{\mathbb{B}} \cup \mathbb{B}\setminus\{x_i\}$. In Table 4, we can see that even if we add the patch-based negative augmentation on top of this stronger contrastive baseline, we can still consistently achieve extra improvement across robustness benchmarks. This shows our proposed patch-based negative augmentation is also complementary to batch-based negative examples in contrastive loss in improving models' robustness. Similar trend holds on other settings, see Table 14 in Appendix. Comparing to the other two negative loss regularizations, uniform and $\ell_2$ loss, we suggest readers to use our proposed contrastive loss in practice since it incorporates the extra benefit of constraining the embeddings of positive pairs to be similar and consistently performs better.

### 5.4 Robustness improvements even under larger pre-training datasets

We further investigate if our proposed method can scale up to larger datasets and continues to be necessary and valuable. To this end, we test if our proposed patch-based negative augmentation still helps robustness when models are pre-trained on ImageNet-21k (10x larger than ImageNet-1k), where the baseline performance is higher. Take P-Shuffle as an example, we display the results in Table 5 with negative augmentation in both pre-training and fine-tuning stages. We see that even when the pre-training dataset is significantly increased, our patch-based negative augmentation can still further improve the robustness of ViT. This demonstrates that our approach is valuable at scale and improves models' robustness from an angle orthogonal to larger pre-training data.

**Mean and standard deviation**: Considering training models from scratch on ImageNet-1k or 21k is very costly, we take the strongest baseline as an example to independently run multiple times. As shown in Table 5, the small standard deviations suggests the improvement is statistically significant. In addition, we have presented **20+** different experimental settings in the paper where our proposed patch-based negative examples consistently improves robustness.

## 5.5 Patch-based negative augmentation reduces texture bias

Geirhos et al. (2018) observed that unlike humans, CNNs rely on more local information (e.g., texture) rather than more global information (e.g., shape) to make a classification. Since our patch-based transformations largely destroy the global structure (e.g., shape), we want to investigate if the non-robust features surviving patch-based transformation overlap with local texture biases. To this end, we evaluate ViT-B/16 trained on patch-based transformations on Conflict Stimuli benchmark (Geirhos et al., 2018), and we see that ViTs trained *only* on patch-based transformation have a 4.9pp to 31.1pp increase on texture bias (Figure 5 in Appendix). This suggests that the useful but non-robust features preserved in patch-based transformation are indeed overlapped with the local texture bias. In addition, using our patch-negative augmentation can also to some extent reduce models' reliance on local texture bias, e.g., we decrease the texture accuracy from 71.7% to 66.5% for ViT-B/16 (Table 6 in Appendix).

## 6 Discussions

**Does ViT become more robust w.r.t. transformed images?** We further evaluate ViTs trained with our robust training algorithms on the patch-based transformed images. We found all three losses on negative views can successfully reduce the prediction accuracy of ViTs to be close to random guess with the original semantic classes as the ground-truth. In other words, our robust training algorithms make ViTs behave similarly as humans on those patch-based transformed images.

**Are high confidence predictions w.r.t. patch transformed images due to non-overlapping patch embeddings?** To answer this, we test original ViT from Dosovitskiy et al. (2021) on the patch-transformed images where the patch size of the patch-based transformations $ps_t$ is not perfectly aligned with the input patch size $ps_i$, e.g., $ps_t$ is not a multiple of $ps_i$. This can approximate the scenario when patch embeddings are extracted from overlapped patches. As shown in Table 11 in Appendix, when the patch size of the patch-based transformation ($ps_t = 24$) and the input patch size ($ps_i = 16$) of ViT are not perfectly aligned, the test accuracy on the patch-based shuffled images decreases from 84.1% to 32.9%. This is lower than the case that the patch sizes are aligned, e.g., the test accuracy is 57.8% when $ps_t = 32$, but still significantly higher than humans (a random guess is close to 0.1%) as well as comparable CNN-based network 9.2%. Therefore, we conjecture the high confidence w.r.t. transformed images is *partially* from non-overlapping patch embedding in ViTs.

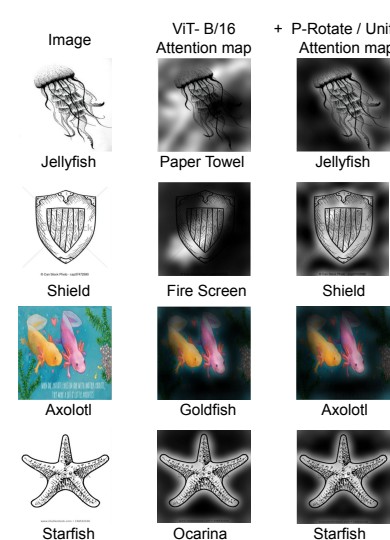

Figure 4: Attention visualization of images misclassified by ViT-B/16 (middle column) and correctly classified by patch-based negative augmentation (P-Rotate) using uniform loss (third column) on ImageNet-R. The ground truth (first column) or the predicted class (second and third column) are displayed at the bottom of each image.

**When does patch-based negative augmentation help?** To answer this question, we follow (Dosovitskiy et al., 2021) to visualize the attention maps of models trained with/without our proposed negative data augmentation. In Figure 4, we can see that our negative augmentation can effectively 1) help models attend to the object to make a correct prediction whereas the standard ViT fails (top two rows), and 2) mitigate models' reliance on unrobust biases, e.g., local bias, color bias, texture bias, etc. For example, the cartoon *Axolotl* (third row) is misclassified by the standard ViT as *Goldfish* due to color similarity and the *Starfish* (bottom row) is incorrectly recognized as *Ocarina* because of local resemblance. This is aligned with other signals in previous sections that standard ViT relies on unrobust biases to make a prediction, whereas our negative augmentation effectively encourages the model to rely less on these unrobust features. Interestingly, for the bottom two rows, the standard ViT has successfully attended to the object in both cases but still failed to make correct predictions, indicating that the attention on the object does not ensure a model is relying on semantically meaningful and robust features to make a prediction.

# 7 Related Work

Vision transformers (Dosovitskiy et al., 2021; Touvron et al., 2021) are a family of Transformer models (Vaswani et al., 2017) that directly process visual tokens constructed from image patch embedding. Unlike convolutional neural networks (LeCun et al., 1989; Krizhevsky et al., 2012; He et al., 2016) that assume locality and translation invariance in their architectures, vision transformers have no such assumptions and can exchange information globally. A few recent studies find pretrained vision transformers are at least as robust as the ResNet counterparts (Bhojanapalli et al., 2021), and possibly more robust (Naseer et al., 2021; Paul & Chen, 2021). Our work studies a specific aspect of robustness pertaining patch-based visual tokens in ViT, and show it may lead to a generalization gap. Different from (Naseer et al., 2021) which also shows ViTs are insensitive to patch operations such as shuffle and occlusion, we design different types of patch-based transformations and develop deeper understanding that the features preserved in patch-based transformations are non-robust. Further, we propose a mitigation strategy based on these patch-based transformations to increase robustness of vision transformers. Another orthogonal line of improving robustness of vision transformers is to develop a deeper understanding of the self-attention mechanisms in vision transformers (Gu et al., 2022) and further advances the neural network architectures (Zhou et al., 2022).

Data augmentation is widely used in computer vision models to improve model performance (Howard, 2013; Szegedy et al., 2015; Cubuk et al., 2020, 2019; Touvron et al., 2021). However, most of the existing data augmentations are "positive" in the sense they assume the class semantic being preserved after the transformation. In this work, we explore "negative" data augmentation operations based on patches, where we encourage the representations of transformed example to be *different* from the original ones. Most related to our work in this direction is the work of Sinha et al. (2020). Although the concept of negative augmentation was proposed in their work, they only apply it for generative and unsupervised modeling without consideration of robustness. In contrast, our work focuses on discriminative and supervised modeling, and demonstrate how such negative examples can reveal specific robustness issues and such augmentation approaches can offer robustness improvements under large-scale pretraining settings.

Our work is also related to contrastive learning (Wu et al., 2018; Hjelm et al., 2019; Oord et al., 2018; He et al., 2020; Tian et al., 2020). The increasing number of negative pairs has shown to be important for representation learning in self-supervised contrastive learning (Chen et al., 2020a), where different images serve as negative examples for each other, and supervised contrastive learning (Khosla et al., 2020), where images with different classes are used as negative examples. Unlike the traditional setting of representation learning, our proposed contrastive loss serves as a regularization term with patch-based negative augmentations as extra negative data points.

# 8 Conclusion

Through this research we have found concrete evidences of ViTs relying on non-robust features to make predictions and shown that this reliance is limiting out-of-distribution robustness. This opens multiple exciting new lines of research. First, our methodology for analyzing the robustness of ViTs provides a valuable recipe for future research. Through designing patch-wise, semantic-destroying transformations that ViTs are insensitive to, we identified which non-robust features models rely on. Second, through negative augmentations during training we reduced the ViTs' reliance on such non-robust features, and improved the out-of-distribution performance of ViTs significantly, without harming in-distribution accuracy. This approach shows the potential for further improving the robustness of ViTs. While we have identified multiple such non-robust features in ViTs, we believe discovering and addressing more is a promising direction for continued progress toward robust vision transformers and vision models in general.

## Acknowledgements

The authors would like to thank Alexey Dosovitskiy, Lucas Beyer and Neil Houlsby for helpful discussions on vision transformer systems. We also want to thank the reviewers for their useful comments.

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
