# Appendix

## A  Patch-based Negative Data Augmentation Reduces Texture Bias

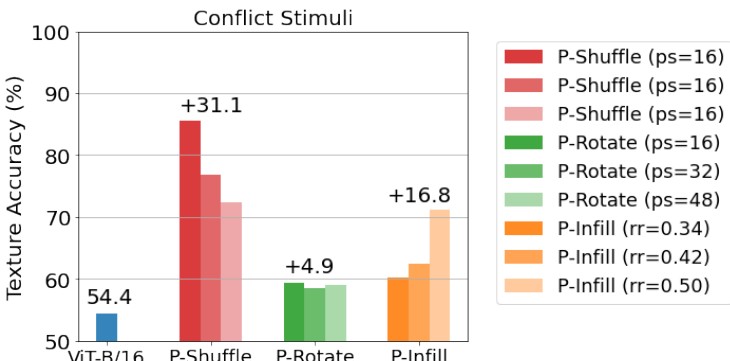

Figure 5: ViTs trained *only* on our patch-based transformations exhibit stronger texture bias. Each bar is the texture accuracy (%) on Conflict Stimuli (Geirhos et al., 2018), and a higher texture accuracy indicates the model has a higher bias towards texture. The "texture accuracy" is defined as the percentage of images that are classified as the "texture" label, provided the image is classified as either "texture" or "shape" label. The baseline model is ViT-B/16 in (Dosovitskiy et al., 2021) trained on original images. Other models are trained on patch-based transformed images, e.g., "P-Shuffle" stands for a ViT-B/16 model trained on patch-based shuffled images. Numbers above the bars are either accuracy (e.g., ViT-B/16) or the *max* accuracy difference between each model family and the baseline ViT-B/16. The patch size in P-Shuffle and P-Rotate and replacement ratio in P-Infill is denoted by "ps" and "rr" respectively.

Table 6: Patch-based negative augmentation effectively reduce models' texture bias on Conflict Stimuli (Geirhos et al., 2018). A higher texture accuracy indicates the model has a higher bias towards texture. The "texture accuracy" is defined as the percentage of images that are classified as the "texture" label, provided the image is classified as either "texture" or "shape" label.

| Pre-train on ImageNet-1k | | Pre-train on ImageNet-21k | |
|---|---|---|---|
| Model | Texture Accuracy | Model | Texture Accuracy |
| ViT-B/16 | 71.7 | Rand-Augment | 57.5 |
| + P-Rotate / Uniform | 66.5 | + P-Shuffle / Uniform | 56.4 |
| + P-Rotate / L2 | 67.2 | + P-Shuffle / L2 | 54.7 |

## B  Training Details

We follow (Dosovitskiy et al., 2021) to train each model using Adam (Kingma & Ba, 2015) optimizer with $\beta_1 = 0.9$, $\beta_2 = 0.999$ for pre-training and SGD with momentum for fine-tuning. The batch size is set to be 4096 for pre-training and 512 for fine-tuning. All models are trained with 300 epochs on ImageNet-1k and 90 epochs on ImageNet-21k in the pre-training stage. In the fine-tuning stage, all models are trained with 20k steps except the models pretrained from ImageNet-1k without Rand-Augment (Cubuk et al., 2020) or Augmix (Hendrycks et al., 2020), which we train them with 8k steps. The learning rate warm-up is set to be 10k steps. Dropout is used for both pre-training and fine-tuning with dropout rate 0.1. If the training dataset is ImageNet-1k, we additionally apply gradient clipping at global norm 1.

Table 7: Training details following (Dosovitskiy et al., 2021).

| Pre-train Dataset | Stage | Base LR | LR Decay | Weight Decay | Label Smoothing |
|---|---|---|---|---|---|
| ImageNet-1K | Pre-train | $3 \cdot 10^{-3}$ | 'cosine' | None | $10^{-4}$ |
| ImageNet-21k | Pre-train | $10^{-3}$ | 'linear' | 0.03 | $10^{-4}$ |
| ImageNet-1K | Fine-tune | 0.01 | 'cosine' | None | None |
| ImageNet-21K | Fine-tune | 0.03 | 'cosine' | None | None |

Table 8: Models using a different hyperparameter $\lambda$ than the default value (1.5).

| Model | Pre-train Dataset | Training stage | Hyperparameter $\lambda$ |
|---|---|---|---|
| Rand-Augment + P-Shuffle / Uniform | ImageNet-1k | Pre-train | 1.0 |
| Rand-Augment + P-Shuffle / Contrastive | ImageNet-1k | Pre-train | 1.0 |
| AugMix + P-Shuffle / L2 | ImageNet-1k | Pre-train | 1.0 |
| AugMix + P-Rotate / L2 | ImageNet-1k | Pre-train | 1.0 |
| AugMix + P-Infill / L2 | ImageNet-1k | Pre-train | 1.0 |
| AugMix + P-Shuffle / Contrastive | ImageNet-1k | Pre-train | 1.0 |
| Rand-Augment + P-Shuffle / Uniform | ImageNet-21k | Pre-train | 0.5 |
| Rand-Augment + P-Shuffle / L2 | ImageNet-21k | Pre-train | 0.5 |
| Rand-Augment + P-Shuffle / Contrastive | ImageNet-21k | Pre-train | 0.5 |
| Rand-Augment + P-Rotate / Uniform | ImageNet-1k | Fine-tune | 0.5 |
| Rand-Augment + P-Infill / Uniform | ImageNet-1k | Fine-tune | 1.0 |
| AugMix + P-Rotate / Uniform | ImageNet-1k | Fine-tune | 1.0 |
| Rand-Augment + P-Shuffle / Uniform | ImageNet-21k | Fine-tune | 0.5 |

## C  Hyper-parameters in Patch-based Negative Augmentation

For the temperature $\tau$ used in contrastive loss, we consistently observe that $\tau = 0.5$ works better in pre-training stage and $\tau = 0.1$ works better in fine-tuning stage. Therefore, we keep this setting for all the models in our paper.

Since we sweep the coefficient $\lambda$ in Eqn. 1 from the set $\{0.5, 1.0, 1.5\}$, we observe that for most of the cases, $\lambda = 1.5$ works the best. In total we have 48 models using loss regularization on negative views in Table 1, Table 2, Table 5 and Table 3. We use $\lambda = 1.5$ for all of them except those listed in Table 8, where either $\lambda = 0.5$ or $\lambda = 1.0$ works better. Actually, we find our proposed negative augmentation is relatively robust to $\lambda$. Therefore, we suggest using $\lambda = 1.5$ if readers do not want to sweep for the best value for this hyperparameter.

In Table. 9, we display the hyperparameters in each patch-based transformation that we use for the reported results in this work. Our algorithms are generally insensitive to these parameters, and we use the same hyperparameter for all the settings investigated in this work.

## D  Ablation Study

### D.1  Sensitivity analysis

We test the sensitivity of our patch-based negative augmentation to various patch sizes in P-Shuffle and P-Rotate, and different replace rates in P-Infill. We find that P-Shuffle and P-Rotate are insensitive to patch sizes from $\{16, 32, 48, 64, 96\}$ for ViT-B/16, and P-Infill is robust to replace rates ranging from 1/3 to 1/2. The accuracy difference is smaller than $0.5\%$ on ImageNet-1k as well as ImageNet-A and ImageNet-R. Therefore, we use the same parameter for all the settings investigated in this work (see Table 9 and Appendix C for details).

### D.2  Double batch-size of baselines

As we use the negative augmented view per example, the effective batch size is doubled compared to the vanilla ViT-B/16 trained with only cross-entropy loss. Therefore, we further investigate if the robustness improvement is a result from a larger batch size. When we increase the batch size from 4096 to 8192 in pre-training while keeping the same 300 training epochs, it decreases the in-distribution accuracy to $76.0\%$ on ImageNet-1k as well as the accuracy on robustness benchmarks,

Table 9: Hyperparameters in patch-based transformations.

| Image Size | Stage | Model | Transformation | Hyperparameter |
|---|---|---|---|---|
| $224 \times 224$ | Pre-train | ViT-B/32 | P-Rotate | patch size = 32 |
| $224 \times 224$ | Pre-train | ViT-B/16 | P-Shuffle | patch size = 32 |
| $224 \times 224$ | Pre-train | ViT-B/16 | P-Rotate | patch size = 16 |
| $224 \times 224$ | Pre-train | ViT-B/16 | P-Infill | replace rate = 15/49 |
| $384 \times 384$ | Fine-tune | ViT-B/32 | P-Rotate | patch size = 64 |
| $384 \times 384$ | Fine-tune | ViT-B/16 | P-Shuffle | patch size = 64 |
| $384 \times 384$ | Fine-tune | ViT-B/16 | P-Rotate | patch size = 32 |
| $384 \times 384$ | Fine-tune | ViT-B/16 | P-Infill | replace rate = 3/8 |

Table 10: P-Shuffle improve accuracy on CIFAR-100 and CIFAR-100-C when ViT-B/16 is pre-trained on ImageNet-21k and then fine-tuned on CIFAR-100.

| Model | CIFAR-100 | CIFAR100-C |
|---|---|---|
| ViT-B/16 | 91.8 | 74.6 |
| + P-Shuffle / Uniform | 92.6 (+0.8) | 77.0 (+2.4) |

e.g., ImageNet-R from 20.3% to 19.3%. Hence we conclude the robustness improvement is from the negative data augmentation.

### D.3 Pre-training vs. Fine-tuning

We further disentangle the effect of patch-based negative data augmentation in pre-training and fine-tuning. Take P-Shuffle as an example, we design experiments to apply negative augmentation 1) only at the fine-tuning stage, 2) only at the pre-training stage, and 3) at both stages. As shown in Table 13, compared to the baselines, patch-based negative augmentation can effectively help improve robustness in both stages, and its effect in pre-training is slightly larger than in fine-tuning. Finally, we found using negative augmentation in both stages during training yields the largest gain.

### D.4 When to use positive data augmentation

As Steiner et al. (2021) observed that traditional (positive) augmentation can slightly hurt the accuracy of ViT if applied to fine-tuning stage, we compare the accuracy of a ViT-B/16 when positive augmentation (e.g., Rand-Augment (Cubuk et al., 2020)) is only applied to pre-training stage as well as both stages. As shown in Table 15, fine-tuning without Rand-Augment achieves slightly better performance.

## E   Visualization of Patch-based Transformations

We display more examples with patch-based transformations without cherry-picking in Figure 6, Figure 7 and Figure 8.

Table 11: Test accuracy on P-Shuffled images with different patch sizes $ps_t$ and clean ImageNet-1k.

| Model | $ps_t$=24 | $ps_t$=32 | ImageNet-1k |
|---|---|---|---|
| ViT-B/16 ($ps_i = 16$) | 32.9 | 57.8 | 84.1 |
| BiT-ResNet101-3 | 9.2 | 24.6 | 84.0 |

Table 12: Effects of patch-based negative augmentation on five different levels of corruption severities on ImageNet-C. Best results are highlighted in **bold**.

| Model | ImageNet-C Corruption Severity Level | | | | |
|---|---|---|---|---|---|
| | 1 | 2 | 3 | 4 | 5 |
| Rand-Augment (Cubuk et al., 2020) | 70.4 | 63.7 | 57.9 | 48.2 | 36.1 |
| + P-Shuffle / Uniform | 71.0 (+0.6) | 64.4 (+0.7) | 59.0 (+1.1) | 49.5 (+1.3) | 37.3 (+1.2) |
| + P-Rotate / Uniform | **71.1 (+0.7)** | 64.6 (+0.9) | 59.0 (+1.1) | 50.0 (+1.8) | 37.6 (+1.5) |
| + P-Infill / Uniform | **71.1 (+0.7)** | 64.6 (+0.9) | 59.1 (+1.2) | 49.5 (+1.3) | 37.3 (+1.2) |
| + P-Shuffle / L2 | 70.5 (+0.1) | 63.9 (+0.2) | 58.3 (+0.4) | 48.6 (+0.4) | 36.6 (+0.5) |
| + P-Rotate / L2 | **71.1 (+0.7)** | **64.8 (+1.1)** | **59.5 (+1.6)** | **50.1 (+1.9)** | **37.8 (+1.7)** |
| + P-Infill / L2 | 70.5 (+0.1) | 63.8 (+0.1) | 58.2 (+0.3) | 48.4 (+0.2) | 36.0 (-0.1) |
| AugMix (Hendrycks et al., 2020) | 71.4 | 65.2 | 60.5 | 51.9 | 40.2 |
| + P-Shuffle / Uniform | 71.6 (+0.2) | 65.7 (+0.5) | 61.2 (+0.7) | 52.8 (+0.9) | 41.4 (+1.2) |
| + P-Rotate / Uniform | 71.7 (+0.3) | 65.7 (+0.5) | 61.1 (+0.6) | 52.7 (+0.8) | 41.4 (+1.2) |
| + P-Infill / Uniform | **71.9 (+0.5)** | 65.8 (+0.6) | 61.1 (+0.6) | 52.4 (+0.5) | 40.8 (+0.6) |
| + P-Shuffle / L2 | 71.8 (+0.4) | 65.8 (+0.6) | 61.0 (+0.5) | 52.4 (+0.5) | 40.7(+0.5) |
| + P-Rotate / L2 | **71.9 (+0.5)** | **66.0 (+0.8)** | **61.5 (+1.0)** | **52.9 (+1.0)** | **41.6 (+1.4)** |
| + P-Infill / L2 | 71.8 (+0.4) | 65.8 (+0.6) | 61.3 (+0.8) | 52.7 (+0.8) | 41.0 (+0.8) |

Table 13: Effect of patch-based negative augmentation in pre-training and fine-tuning stages. Top-1 accuracies of ViT-B/16 pretrained and fine-tuned on ImageNet-1k. Under 'Stage' we denote which training stage patch-based negative augmentation is used. The best result under each setting is highlighted in **bold**.

| | Pre-train on ImageNet-1k | | | | |
|---|---|---|---|---|---|
| Model | Stage | ImageNet-1k | ImageNet-A | ImageNet-C | ImageNet-R |
| Rand-Augment (Cubuk et al., 2020) | - | 79.1 | 7.2 | 55.2 | 23.0 |
| + P-Shuffle / Uniform | Fine-tune | 79.1 | 7.1 | 55.3 | 23.0 |
| + P-Shuffle / Uniform | Pre-train | 79.3 | 7.6 | 56.2 | **23.5** |
| + P-Shuffle / Uniform | Both | **79.3** | **7.7** | **56.2** | 23.4 |
| + P-Shuffle / Contrastive | Fine-tune | 79.5 | 7.6 | 56.2 | 23.7 |
| + P-Shuffle / Contrastive | Pre-train | 79.4 | 8.5 | 56.8 | 24.0 |
| + P-Shuffle / Contrastive | Both | **79.7** | **8.9** | **57.8** | **24.7** |
| | Pre-train on ImageNet-21k | | | | |
| Model | Stage | ImageNet-1k | ImageNet-A | ImageNet-C | ImageNet-R |
| Rand-Augment (Cubuk et al., 2020) | - | 84.4 | 28.7 | 67.2 | **38.7** |
| + P-Shuffle / L2 | Fine-tune | 84.5 | 29.4 | 67.9 | 39.0 |
| + P-Shuffle / L2 | Pre-train | 84.4 | **29.9** | 67.5 | 38.8 |
| + P-Shuffle / L2 | Both | **84.5** | 29.7 | **68.0** | **39.6** |
| + P-Shuffle / Contrastive | Fine-tune | 84.4 | 29.2 | 67.5 | **38.7** |
| + P-Shuffle / Contrastive | Pre-train | **84.6** | 29.9 | 67.7 | 38.5 |
| + P-Shuffle / Contrastive | Both | 84.3 | **30.8** | **68.1** | 38.6 |

Table 14: Effect of patch-based negative augmentation in contrastive loss regularization. Top-1 accuracies of ViT-B/16 trained with or without patch-based negative augmentation.

| Pre-train on ImageNet-1k | | | | |
|---|---|---|---|---|
| Model | ImageNet-1k | ImageNet-A | ImageNet-C | ImageNet-R |
| ViT-B/16 + Contrastive* | 78.7 | 8.1 | 53.5 | 22.8 |
| ViT-B/16 + Shuffle / Contrastive | 78.9 | 8.2 | 54.1 | 23.2 |
| ViT-B/16 + P-Rotate / Contrastive | 78.9 | 8.6 | 54.1 | 23.6 |
| Rand-Augment + Contrastive* | 79.7 | 8.9 | 57.6 | 24.7 |
| Rand-Augment + P-Rotate / Contrastive | 79.9 | 9.4 | 58.4 | 25.4 |
| Rand-Augment + P-Infill / Contrastive | 79.9 | 9.3 | 57.9 | 25.0 |
| AugMix + Contrastive* | 79.6 | 9.0 | 59.8 | 27.2 |
| AugMix + P-Rotate / Contrastive | 79.6 | 9.8 | 60.0 | 27.5 |
| AugMix + P-Infill / Contrastive | 79.6 | 9.9 | 60.3 | 27.3 |
| Pre-train on ImageNet-21k | | | | |
| Model | ImageNet-1k | ImageNet-A | ImageNet-C | ImageNet-R |
| Rand-Augment + Contrastive* | 84.1 | 29.7 | 67.6 | 39.2 |
| Rand-Augment + P-Shuffle / Contrastive | 84.3 | 30.8 | 68.1 | 38.6 |

Table 15: Effect of positive augmentation in pre-training and fine-tuning stages. Top-1 accuracies of ViT-B/16 pretrained on ImageNet-21k and fine-tuned on ImageNet-1k. Under 'Stage' we denote which training stage Rand-Augment (Cubuk et al., 2020) is used.

| Model | Stage | ImageNet-1k | ImageNet-A | ImageNet-C | ImageNet-R |
|---|---|---|---|---|---|
| Rand-Augment | Pre-train | 84.4 | 28.7 | 67.2 | 38.7 |
| Rand-Augment | Both | 84.4 | 29.1 | 67.0 | 38.4 |

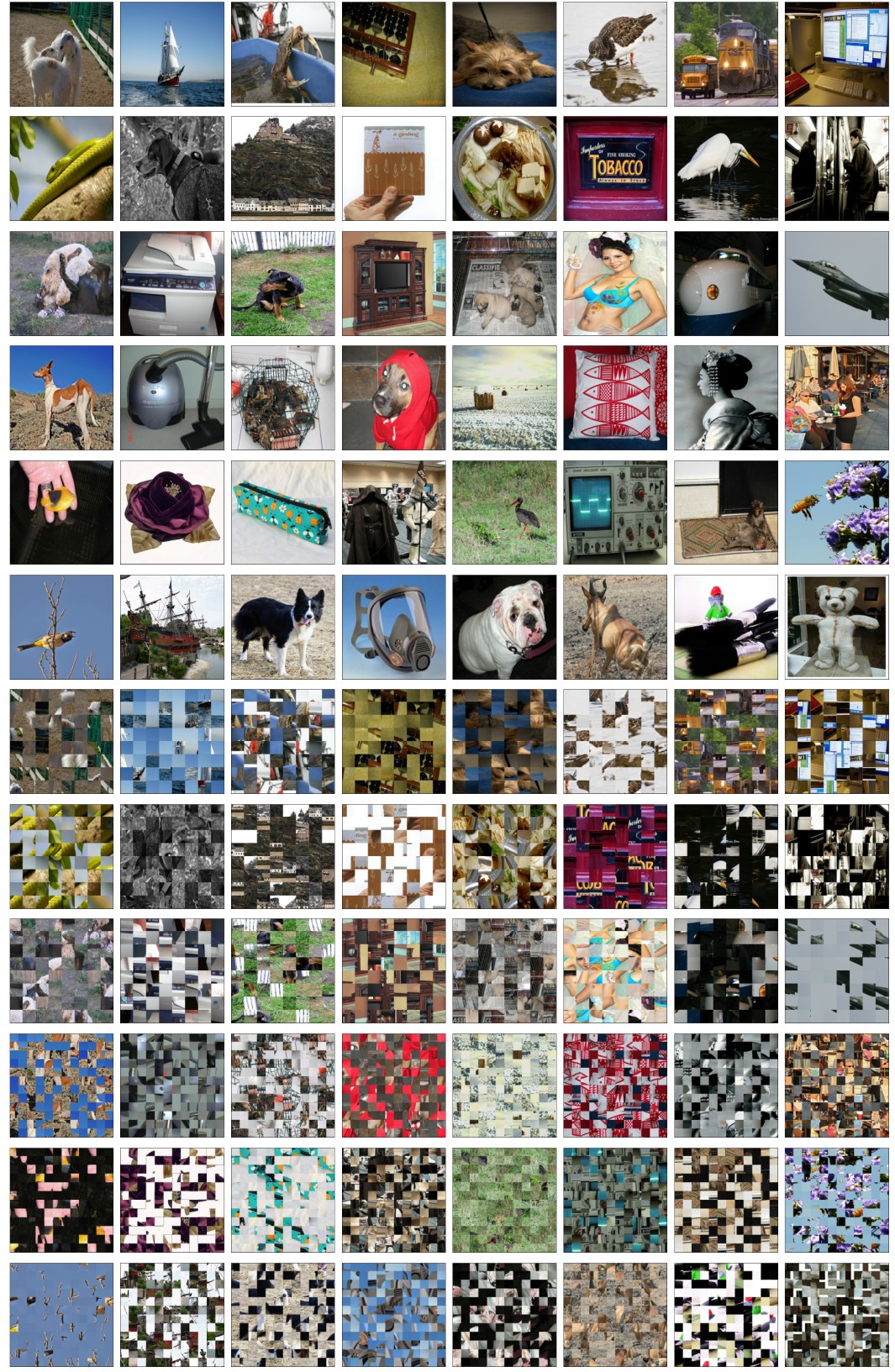

Figure 6: Examples of original images (on the top) and their corresponding patch-based shuffle (at the bottom) with either patch size 32 or 48 without cherry-picking.

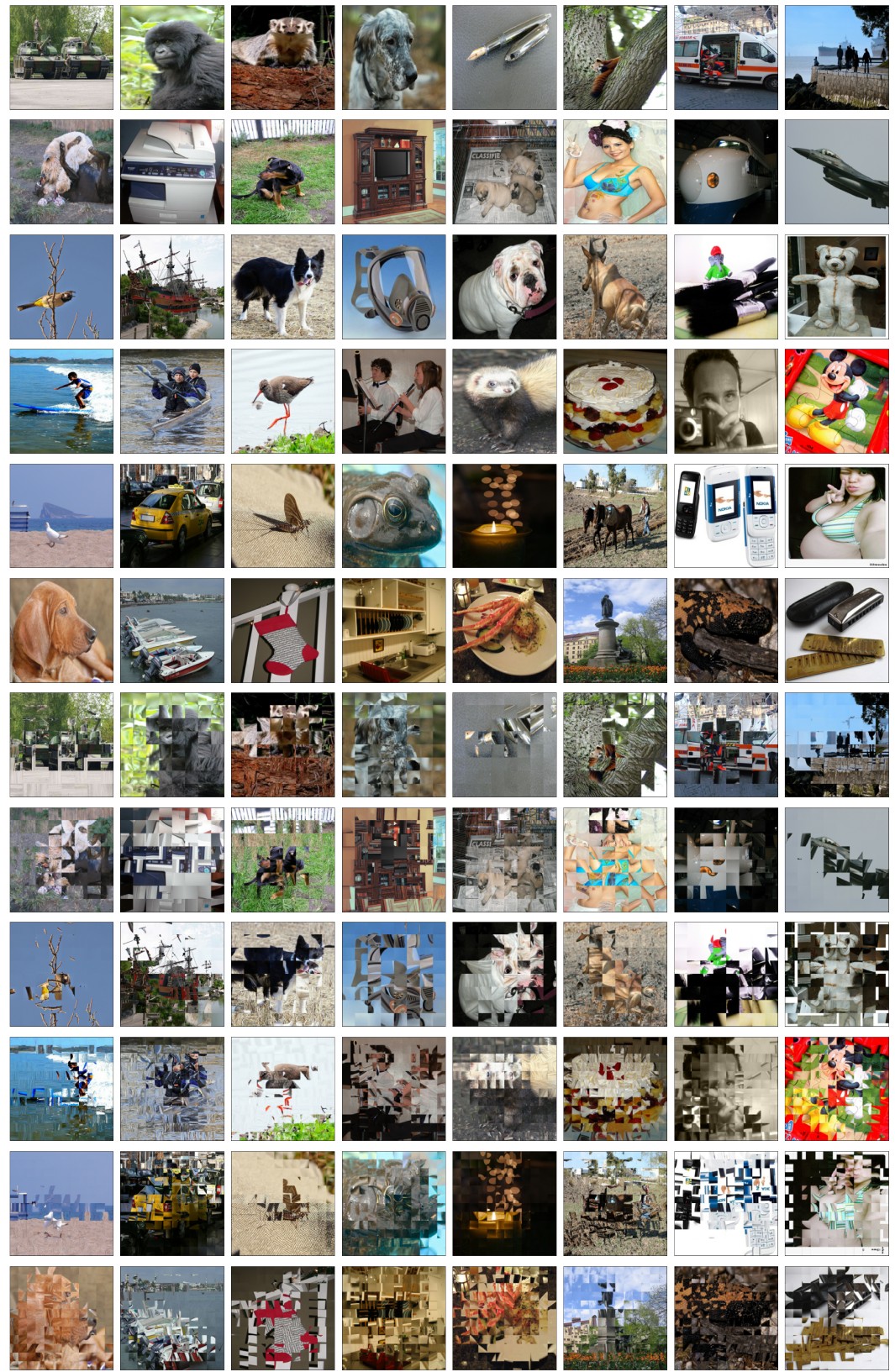

Figure 7: Examples of original images (on the top) and their corresponding patch-based rotation (at the bottom) with either patch size 32 or 48 without cherry-picking.

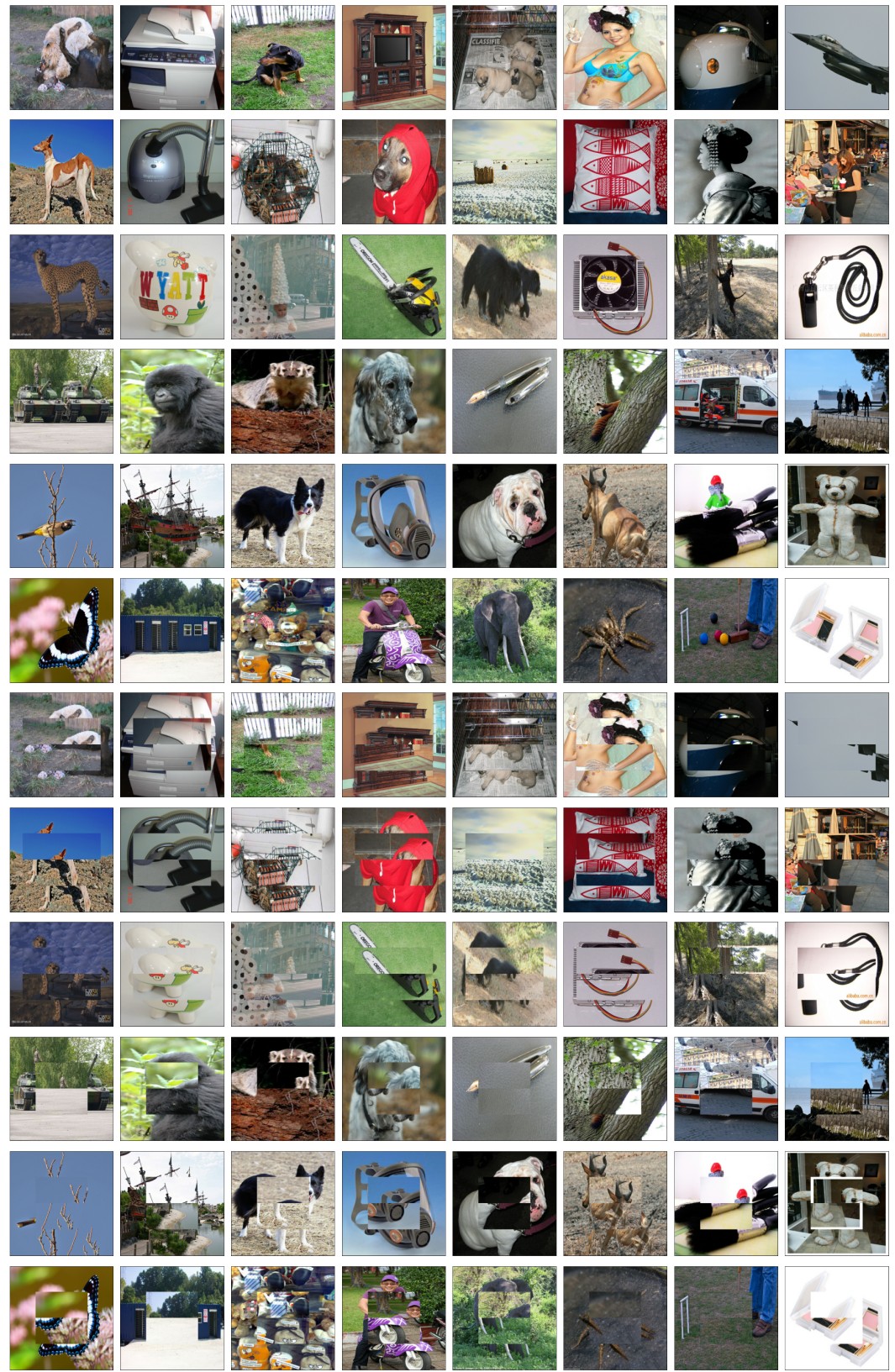

Figure 8: Examples of original images (on the top) and their corresponding patch-based infill (at the bottom) with either replace rate 0.25 or 0.375 without cherry-picking.