# OpenReview forum: "Understanding and Improving Robustness of Vision Transformers through Patch-based Negative Augmentation"
_NeurIPS.cc/2022/Conference — NeurIPS 2022 Accept_

### Official Review · Reviewer_9VjA · 2022-07-11

**Rating:** 7
**Confidence:** 4
**Soundness:** 4 excellent
**Presentation:** 4 excellent
**Contribution:** 3 good

**Summary:**

This paper studies the robustness of ViT models. The authors find that ViT models can keep high in-distribution accuracy even when training on images with heavy patch-based transformations such as shuffle, rotate, and infill, which indicates that ViTs rely on the features that survived these transformations. However, these features are non-robust and render a significant accuracy drop on the out-of-distribution datasets. Based on such observation, the authors propose negative augmentation to improve the model robustness against data distribution shift. The algorithm first generates negatively augmented training samples using the patch-based transformation. It then incorporates the Uniform loss and L2 loss to regularize the training away from learning these non-robust features. The authors conduct extensive experiments to demonstrate the effectiveness of the proposed approach. The negative augmentation is complementary to existing positive augmentation such as AugMix and batch-based negative examples in contrastive loss, and can further improve the robustness on top of that.

**Questions:**

1. In equation 1, for the loss on the negative augmented training samples, is it L_{neg}(B, \tilde(B); \theta) or L_{neg}(\tilde(B); \theta)? Do you use the clean dataset twice in the training? If so, what is the reason for doing so?

2. For the experiments that use Uniform loss, how will the performance change if you use the regular cross-entropy loss on the negative augmented examples?

3. In Table 1, Uniform loss achieves the best performance on ViT-B/16 while the L2 loss achieves the best performance on ViT-B/32, what is the reason behind this? In practice, how do we decide which loss to use? Also, is there any strategy to pick the best type of patch-based transformation for negative augmentation?

4. Did you try to train models with the combination of Uniform loss and L2 loss, will that further improve the robustness or undermine the robustness?

5. Can you please explain the setting difference between lines 306-307 and lines 133-135?

6. Another related direction to enhance model robustness is to improve the model architecture (e.g. Daquan Zhou et.al, Understanding The Robustness in Vision Transformers). It would be desirable if the authors could discuss this orthogonal research direction in the related work.

7. A small typo in line 35: showingthat --> showing that

**Limitations:**

Please refer to the questions above.

**Strengths And Weaknesses:**

Strengths:
1. This paper studies an important topic --  model robustness against OOD data, and proposes an effective approach to improve the robustness.
2. The authors demonstrate an intriguing property of ViTs that they are resilient to patch-based transformations. They further show that the features that survived such transformations are not robust, which is a novel discovery.
3. Based on the observation they propose negative augmentation and two losses to train models away from using the non-robust features. The proposed approach is technically sound and easy to understand.
4. The authors perform extensive experiments to evaluate the effectiveness of the proposed method, the results are convincing and support their claims.
5. The authors also show that the proposed method can collaborate with positive data augmentation and batch-based negative examples in contrastive loss and further improve the robustness.
6. The paper is well-organized and easy to follow.

Weakness:
1. For some experiment results, the authors didn't explicitly explain the rationale and possible reasons. Please refer to the Questions for details.
2. Missing discussion about related works in the same field. Please refer to the Questions for details.

---

> ### Author Response · Authors · 2022-08-02
> **Response to Reviewer 9VjA**
>
> Thanks very much for your great support of our work and we would love to clarify your questions as follows.
>
> Q1: In equation 1, for the loss on the negative augmented training samples, is it $L_{neg}(B, \tilde(B); \theta)$ or $L_{neg}(\tilde(B); \theta)$? Do you use the clean dataset twice in the training? If so, what is the reason for doing so?
>
> A1: For uniform loss, it is $L_{neg}(\tilde(B); \theta)$ since we only need to apply uniform labels for negative examples. However, for L2 loss, we need the clean data information since we need to encourage the prediction between clean and negative examples to be far away. Therefore, the notation should be $L_{neg}(B, \tilde(B); \theta)$. We do not use the clean dataset twice and will make it more clear in the revised version.
>
> Q2: For the experiments that use Uniform loss, how will the performance change if you use the regular cross-entropy loss on the negative augmented examples?
>
> A2: When we use uniform loss, the negative regularization is still regular cross-entropy loss on the negative augmented examples, but for these negative examples we use uniform labels as ground truth rather than one-hot labels.
>
> Q3: Any strategy to pick the loss as well as the best type of patch-based transformation for negative augmentation?
>
> A3: Under multiple scenarios that we investigated, uniform loss achieves similar performance as L2 loss. In practice, we suggest readers to use our proposed contrastive loss with patch-based negative augmentation in Section 5.3 since it incorporates the extra benefit of constraining the embeddings of positive pairs to be similar and consistently performs better.
>
> In terms of negative augmentation, we would suggest readers use a combination of different types of negative transformations, which is similar to the standard way to incorporate different types of positive data augmentation (e.g., RandAug, AugMix).
>
> Q4: Combination of Uniform loss and L2 loss
>
> A4: We did not try to combine Uniform loss and L2 loss empirically. However, intuitively we expect the performance would be similar to the case using only one loss. This is because combining two losses is similar to increasing the hyperparameter $\lambda$ in Eqn. 1, which balances the importance of the negative augmentation. Empirically we observe that the effect of negative examples is stable when we vary the hyperparameter $\lambda$ between [1, 2].
>
> Q5: Can you please explain the setting difference between lines 306-307 and lines 133-135?
>
> A5: Line 306-307: This experimental setting is to show that our proposed negative data augmentation can effectively reduce the model's reliance on features preserved in small patches and perform similarly as humans.  We **train** the model on the **clean images** using our proposed negative data augmentation. We then test the model on the patch-transformed images and find that the model could not identify the semantic class of these patch-transformed images.
>
> Line 133-135: This experimental setting is to show that if the model only captures the features preserved in small patches, the robustness of the model degrades significantly. Specifically, we **train** the model **only on patch-based transformed images** with the original semantic class assigned as their ground-truth.
>
> Q6: Another related direction to enhance model robustness is to improve the model architecture (e.g. Daquan Zhou et.al, Understanding The Robustness in Vision Transformers).
>
> A6: Thanks very much for pointing out this work to us. We will add this work into related work and discuss more about the relationship between model architecture and robustness.
>
> Q7: A small typo in line 35: showingthat --> showing that
>
> A7: Thanks very much for pointing out this to us. We will correct it in the updated version.

---

> > ### Comment · Reviewer_9VjA · 2022-08-05
> > **The authors addressed all my concerns.**
> >
> > I want to thank the authors for the detailed rebuttal, which addressed all my concerns. I would like to keep my rating.

---

### Official Review · Reviewer_5kdm · 2022-07-11

**Rating:** 7
**Confidence:** 4
**Soundness:** 3 good
**Presentation:** 4 excellent
**Contribution:** 3 good

**Summary:**

This paper investigates a small but non-obvious problem: vision transformers (ViTs) are known to be non-sensitive to patch shuffling, in contrast to humans. Could we make ViTs more robust by making them sensitive to patch shuffling (and other patch-based transformations)? The authors performed extensive experiments, and the benefits of doing so is decent and consistent across various settings.

**Questions:**

1. Is the negative augmentation done to every batch, or every N batches?
2. It will help readers understand whether/how the ViT attends to different patterns after applying negative augmentation, using some attribution visualizations.



**Strengths And Weaknesses:**

Strengths:
1. The investigated problem involves a basic aspect of ViT training. The benefits of this method can be cheaply incorporated to various downstream tasks.
2. The experiments are very extensive and convincing.

Weaknesses:
1. Some of the most significant improvements are achieved with patch-rotate (e.g. Table 1). But the insensitivity of vit to patch-rotate seems to be a result of the "whole-image rotation" augmentation (part of RandAug)? This is quite different from the insensitivity of vit to patch-shuffling, which is due to the invariance of transformers to token orders. So I think patch-shuffling is more inherent and more interesting to vits.
2. I'm still not totally sure that the shuffle invariance of ViTs has only negative effects and should be completely eliminated. For example, in MAE [a], ViTs can recognize an image even when most patches are masked. Given this, it's natuaral that when patches are randomly shuffled, ViTs can still recognize the image. Therefore, I'm a bit concerned that maybe there are deeper explanations of the shuffle invariance, instead of just "exploiting texture/local patterns biases".

[a] Masked Autoencoders Are Scalable Vision Learners Kaiming, CVPR 2022.

---

> ### Author Response · Authors · 2022-08-02
> **Response to Reviewer 5kdm**
>
> Thank you very much for your support to our work.  We will try to clarify your questions here and plan to update the paper to clarify these points.
>
> Q: Is the negative augmentation done to every batch, or every N batches?
>
> The negative augmentation is done to every batch.
>
> Q: Attribute visualization.
>
> We will include attribution visualizations to show how models trained with/without negative augmentation attend to different patterns.

---

> > ### Comment · Reviewer_5kdm · 2022-08-08
> > **Weakness 1 unaddressed**
> >
> > Thanks the authors for the clarification. However I do wish to hear some comments on weakness 1, i.e., how important it is to make vit sensitive to patch-shuffling.

---

> > > ### Author Response · Authors · 2022-08-08
> > > **Answer to Weakness 1**
> > >
> > > Thanks very much for the question and we would love to answer your question as follows:
> > >
> > > First, we agree with the reviewer that patch-shuffling is a unique transformation for vision transformers since it employs the invariance to token orders. In the first part of Section 6: Discussions, we validated that vision transformers can no longer recognize patch-shuffled images if we incorporate them as negative examples into training. That means, if we improve the sensitivity of vision transformers against patch-shuffling, we can effectively improve the robustness of out-of-distribution data, supported by the improvement of patch-based shuffling for models pretrained on ImageNet-1k (Table 2) as well as models pretrained on ImageNet-21k (Table 5). Second, we do **not** think the insensitivity of vision transformers to patch-rotation is just a result of whole-image rotation augmentation. This is because 1) All the experiments in “Section 3: Understanding the robustness of ViTs” are based on models trained without whole-image rotation augmentation and we still observe that vision transformers are insensitive to patch-rotation. 2) In Table 1, we do not incorporate whole-image based rotation (no RandAug) and we can still see that patch-rotation can consistently improve out-of-distribution robustness.

---

### Official Review · Reviewer_BLbX · 2022-07-11

**Rating:** 5
**Confidence:** 3
**Soundness:** 4 excellent
**Presentation:** 4 excellent
**Contribution:** 3 good

**Summary:**

In this paper, the authors use the special patch-based data processing pipeline of ViTs to study their robustness. Authors find that ViTs tend to overfit on some unrecognizable features in the patch, making them not robust to distribution shifts. To alleviate this issue, the authors propose to use patch-based negative augmentations to train the ViTs and boost their robustness. Experiments show that the proposed patch-based negative augmentations can boost the robustness of ViTs and are complementary to traditional (positive) data augmentation techniques and batch-based negative examples in contrastive learning.

**Questions:**

- A curious question is whether the observed phenomenon exists in other vision tasks (e.g., detection and segmentation)?

**Limitations:**

The authors have adequately addressed the limitations and potential negative societal impact of their work.

**Strengths And Weaknesses:**

Strengths:
- The observation on overfitting and its reason is interesting.
- The proposed method is simple and effective in boosting the trained ViTs robustness.
- Detailed evaluations are provided to show the performance improvement.
- Writing is clear and easy to follow

Weakness:
- Only evaluated on traditional ViTs, how about more recent ViTs? Will more intense use of convolution layers alleviate such overfitting issues mentioned?

---

> ### Author Response · Authors · 2022-08-02
> **Response to Reviewer BLbX**
>
> Thank you for your review and thoughtful feedback.  We will try to clarify your questions here and plan to update the paper to clarify these points.
>
> Q1: Only evaluated on traditional ViTs, how about more recent ViTs? Will more intense use of convolution layers alleviate such overfitting issues mentioned?
>
> A1: Our work mainly focuses on the traditional ViTs as this is the newest building block for image models and has been comparatively studied much less than CNNs. It would be an interesting future research direction to study if other ViT-based architectures share similar problems. Intuitively, we do not think that more convolutional layers can alleviate the problem because we empirically observe a hybrid architecture (including both convolutional layers as well as ViT) suffer from the same problem.
>
> Q2: A curious question is whether the observed phenomenon exists in other vision tasks (e.g., detection and segmentation)?
>
> A2: This is a really interesting point and we think would make for great future research.  In part, we believe that this is a property of ViT model architecture and try to limit the catering to the particulars of classification tasks, but it is possible that tasks like segmentation are less vulnerable to some of these spurious correlations (but might be more vulnerable to other spurious patterns like always looking for the regions with high-contrast), which would be quite interesting to find out.

---

### Official Review · Reviewer_QKPp · 2022-07-12

**Rating:** 5
**Confidence:** 5
**Soundness:** 4 excellent
**Presentation:** 3 good
**Contribution:** 3 good

**Summary:**

This is an interesting paper that discusses the robustness of the Transformer’s patch-wise architectural structure. The authors experimentally show that patch-based transformations largely destroy images to be unrecognizable to humans, but ViT recognizes them as the original class (e.g., keeshond or magpie) with high confidence as shown in Figure 1. Based on this investigation, they proposed patch-based negative augmentation and experimentally demonstrated that it consistently improves robustness of ViTs on ImageNet based robustness benchmarks across 20+ different experimental conditions.

**Questions:**

Please answer to my comments above.

**Limitations:**

Nothing special.

**Strengths And Weaknesses:**

Strength:
+ This is an interesting paper that discuss its robustness to the patch-wise data destroy. The motivation is very clear.
+ The patch-based negative augmentation based on the above observation is interesting and reasonable.
+ The experimental validation is throughout.

Weakness:
- The performance improvement from the baseline is not very large. Therefore, I admit that the concept itself is interesting, but I wonder how much this negative augmentation would give impacts to the community.
- Although it is claimed that the proposed method outperforms the baseline, the best condition differs from dataset to dataset. The authors should discuss the theoretical background of which approach is better in which case. If the authors could provide a strategy to choose the best setting, that would be great.
- The authors may want to show some specific cases where the proposed method helps a lot, not only discussing the average accuracy.

---

> ### Author Response · Authors · 2022-08-02
> **Response to Reviewer QKPp**
>
> Thank you for your review and thoughtful feedback.  We will try to clarify your questions here and plan to update the paper to clarify these points.
>
> Q1: The performance improvement from the baseline is not very large. Therefore, I admit that the concept itself is interesting, but I wonder how much this negative augmentation would give impacts to the community.
>
> A1: We see our contribution to the community as multi-pronged.  First, we appreciate the reviewer agreeing that one of our major contributions is to discover that ViTs rely on features preserved in small patches and these features are not robust. Second, we demonstrate this insight can meaningfully benefit model development.  In terms of the empirical improvement, since most of our experiments are performed at ImageNet-scale, our methods can consistently have around 1-2 points improvement. Last, while we demonstrate immediate benefits of these insights, we believe our work shows the potential for further improving the robustness of ViTs. While we have identified multiple such non-robust features in ViTs, we believe that discovering and addressing more is an interesting future direction to further improve ViTs.
>
> Q2: Although it is claimed that the proposed method outperforms the baseline, the best condition differs from dataset to dataset. The authors should discuss the theoretical background of which approach is better in which case. If the authors could provide a strategy to choose the best setting, that would be great.
>
> A2: Although the best condition between uniform loss and L2 loss differs from dataset to dataset, there is not a significant difference between these two losses in all the scenarios (20+) that we investigated. In practice, we suggest readers use our proposed contrastive loss with patch-based negative augmentation in Section 5.3 since it incorporates the extra benefit of constraining the embeddings of positive pairs to be similar and consistently performs better than uniform and L2 losses.
>
> Q3: The authors may want to show some specific cases where the proposed method helps a lot, not only discussing the average accuracy.
>
> A3: Actually, we have displayed Table 3 in order to have a clear picture how negative augmentation improves robustness on data with different degrees of corruptions. Specifically, we display the top-1 accuracy on ImageNet-C with five different levels of corruption severity in Table 3. The general trend emerges: as the corruption level goes higher, the benefit of negative examples becomes larger. This suggests our proposed negative examples are especially useful when data is highly corrupted. In addition, we will also include attribution visualizations to show how models trained with/without negative augmentation attend to different patterns in the revised version.

---

> > ### Comment · Reviewer_QKPp · 2022-08-07
> > **About Q3**
> >
> > Thank you for the reply.
> > I am almost satisfied, but have additional comments on Q3.
> > Even in Table 3, only the statistics are shown. I wanted to see some specific cases (images) that can be correctly recognized with your method while the others fail.

---

> > > ### Author Response · Authors · 2022-08-09
> > > **Thanks for your suggestion**
> > >
> > > Thanks very much for your suggestion and we will add a section displaying specific images that can be recognized by our method while others fail. To briefly give you an impression, our negative augmentation can effectively mitigate models' reliance on biases, e.g., texture bias, color bias, local bias, etc. For example, one cartoon **Axolotl** from ImageNet-R, whose color is yellow, is misclassified by standard ViT as **Goldfish** but correctly recognized by our model trained with negative examples. This is aligned with other signals that standard ViT relies on the color bias to make a prediction, whereas our model makes a prediction based on more semantic meaningful features.

---

### Meta-Review · Area_Chair_T5Qn · 2022-08-26

**Recommendation:** Accept
**Confidence:** Certain

**Metareview:**

The reviewers agree that this is an interesting work. Some concerns were expressed, especially regarding the limited gain of the method and the generality to other types of ViTs. But overall the reviewers recognize the technical contribution from this paper and the AC agrees with their decisions.

**Award:**

No

---

### Decision · Program_Chairs · 2022-09-14

Accept